# Histidine phosphorylation relieves copper inhibition in the mammalian potassium channel KCa3.1

**Shekhar Srivastava[1,2,3], Saswati Panda[1,2,3], Zhai Li[1,2,3], Stephen R Fuhs[4], Tony Hunter[4], Dennis J Thiele[5,6], Stevan R Hubbard[1,3]\*, Edward Y Skolnik[1,2,3]\***

[1]Department of Biochemistry and Molecular Pharmacology, New York University School of Medicine, New York, United States; [2]Division of Nephrology, New York University School of Medicine, New York, United States; [3]Kimmel Center for Biology and Medicine, Skirball Institute of Biomolecular Medicine, New York University, New York, United States; [4]Molecular and Cell Biology Laboratory, Salk Institute for Biological Studies, La Jolla, United States; [5]Department of Pharmacology and Cancer Biology, Duke University School of Medicine, Durham, United States; [6]Department of Biochemistry, Duke University School of Medicine, Durham, United States

**Abstract** KCa2.1, KCa2.2, KCa2.3 and KCa3.1 constitute a family of mammalian small- to intermediate-conductance potassium channels that are activated by calcium-calmodulin. KCa3.1 is unique among these four channels in that activation requires, in addition to calcium, phosphorylation of a single histidine residue (His358) in the cytoplasmic region, by nucleoside diphosphate kinase-B (NDPK-B). The mechanism by which KCa3.1 is activated by histidine phosphorylation is unknown. Histidine phosphorylation is well characterized in prokaryotes but poorly understood in eukaryotes. Here, we demonstrate that phosphorylation of His358 activates KCa3.1 by antagonizing copper-mediated inhibition of the channel. Furthermore, we show that activated CD4[+] T cells deficient in intracellular copper exhibit increased KCa3.1 histidine phosphorylation and channel activity, leading to increased calcium flux and cytokine production. These findings reveal a novel regulatory mechanism for a mammalian potassium channel and for T-cell activation, and highlight a unique feature of histidine versus serine/threonine and tyrosine as a regulatory phosphorylation site.

**\*For correspondence:** stevan. hubbard@med.nyu.edu (SRH); edward.skolnik@med.nyu.edu (EYS)

## Introduction

KCa2.1, KCa2.2, KCa2.3 and KCa3.1 (also called SK1–4) are encoded by the *KCNN* genes and respond to calcium via calmodulin, which is constitutively bound to the cytoplasmic region of these channels (*Adelman et al., 2012*). KCa2.1, KCa2.2 and KCa2.3 are expressed predominantly in neurons, contributing to medium afterhyperpolarization, whereas KCa3.1 plays a key role in the activation of T cells, B cells and mast cells (*Feske et al., 2015*). Potassium efflux via KCa3.1 is required to maintain a negative membrane potential, which provides the electrical gradient for sustained calcium influx via calcium release-activated channels (CRACs) and subsequent production of cytokines (*Feske et al., 2015*).

A unique feature of KCa3.1 relative to the other KCa channels is its regulation by histidine phosphorylation. We showed previously that His358 of KCa3.1 is phosphorylated (pHis358) by nucleoside diphosphate kinase-B (NDPK-B) (*Di et al., 2010*; *Srivastava et al., 2006b*), which, along with NDPK-A, are the only two mammalian protein histidine kinases identified to date (*Attwood and Wieland,*

2015). We also showed that KCa3.1 activation requires phosphatidylinositol 3-phosphate (PI(3)P) (*Srivastava et al., 2006a*), generated by a class II phosphatidylinositol 3-kinase (PI3K-C2β) (*Srivastava et al., 2009*), and that KCa3.1 is negatively regulated by protein histidine phosphatase-1 (PHPT1), which dephosphorylates pHis358 (*Srivastava et al., 2008*), and by myotubularin-related protein-6 (MTMR6), which dephosphorylates PI(3)P (*Srivastava et al., 2005*). In addition, we recently identified phosphoglycerate mutase-5 (PGAM5) as a histidine phosphatase that specifically dephosphorylates the catalytic histidine (His118) in NDPK-B. By dephosphorylating NDPK-B, PGAM5 negatively regulates T-cell receptor signaling by inhibiting NDPK-B-mediated histidine phosphorylation and activation of KCa3.1 (*Panda et al., 2016*).

We reported previously that mutation of His358 (H358N) converted KCa3.1 into a channel that, like the other three KCa channels, requires only calcium-calmodulin for activation (*Srivastava et al., 2006b*). Furthermore, swapping 14 residues of KCa3.1 containing His358 with the equivalent residues of KCa2.3 converted the latter into a channel that required NDPK-B and PI(3)P for activation (*Srivastava et al., 2006a*). These studies highlighted the autonomous role of His358 and proximal residues in the regulation of KCa3.1.

Although histidine phosphorylation is well characterized in prokaryotic two-component systems used in chemotaxis and other sensing systems (*Hess et al., 1988*), it is poorly characterized in eukaryotes (*Klumpp and Krieglstein, 2009*), in part because phosphohistidine is more labile than phosphotyrosine or phosphoserine/threonine. In addition to KCa3.1, histidine phosphorylation of several mammalian proteins by NDPKs has been reported, including the β subunit of heterotrimeric G proteins and the transient receptor potential vanilloid-5 (TRPV5) channel (*Attwood and Wieland, 2015*; *Cai et al., 2014*; *Klumpp and Krieglstein, 2009*). However, the functional consequences of histidine phosphorylation of these eukaryotic proteins, and the mechanisms whereby histidine phosphorylation regulates their activity, are poorly understood. The regulation of KCa3.1 by histidine phosphorylation has emerged as the clearest example in a mammalian protein of the functional importance of this post-translational event, yet the molecular basis for His358-mediated regulation of KCa3.1 is unknown.

The special role of histidine in KCa3.1 inhibition, together with the knowledge that histidine is a common ligand in metal-ion coordination, led us to hypothesize that the four copies of His358 in the cytoplasmic domains of the homotetrameric channel coordinate a metal ion, which renders KCa3.1 refractory to the conformational changes induced by calcium binding to calmodulin. Here, we provide evidence for copper-mediated inhibition of KCa3.1 from patch-clamping studies of KCa3.1 in human embryonic kidney (HEK) 293 cells and in mouse embryonic fibroblasts (MEFs) from copper transporter-1 (Ctr1) knockout mice. Moreover, we show that copper inhibition of KCa3.1 is relevant in a physiologic context, namely, regulation of CD4[+] T-cell activation.

## Results

### KCa3.1 is activated by metal chelators and inhibited by copper in whole-cell membrane patches

To test the hypothesis that KCa3.1 is inhibited by His358-mediated metal binding, we first used whole-cell patch clamping to measure the effect of the cell-permeable metal chelator TPEN (N,N,N',N'-tetrakis(2-pyridylmethyl) ethylenediamine) on the current from HEK 293 cells stably transfected with GFP-KCa3.1 (293-KCa3.1 cells). TPEN is a high-affinity chelator ($K_D < 10^{-10}$) of several metal ions, including zinc, copper, iron, nickel and manganese (*Smith and Martell, 1975*; *Treves et al., 1994*). As shown in *Figure 1A,B,E*, the addition of 20 µM TPEN to the cell bath markedly increased the measured current from these cells, consistent with metal-ion inhibition of KCa3.1. We identified the KCa3.1-mediated current as a major contributor to the current from these cells since it was largely blocked by 1 µM TRAM-34, a KCa3.1-specific inhibitor (*Wulff et al., 2000*).

Histidine often serves as a ligand for zinc and copper (*Dokmanić et al., 2008*). We therefore tested whether addition of these ions could reverse the TPEN-mediated activation of KCa3.1. Addition of 100 µM zinc ($ZnCl_2$) did not alter substantially the measured current from 293-KCa3.1 cells (*Figure 1A,E*), but addition of 100 µM copper ($CuCl_2$) to the bath caused a rapid decrease in the current (*Figure 1B,E*), consistent with channel inhibition.

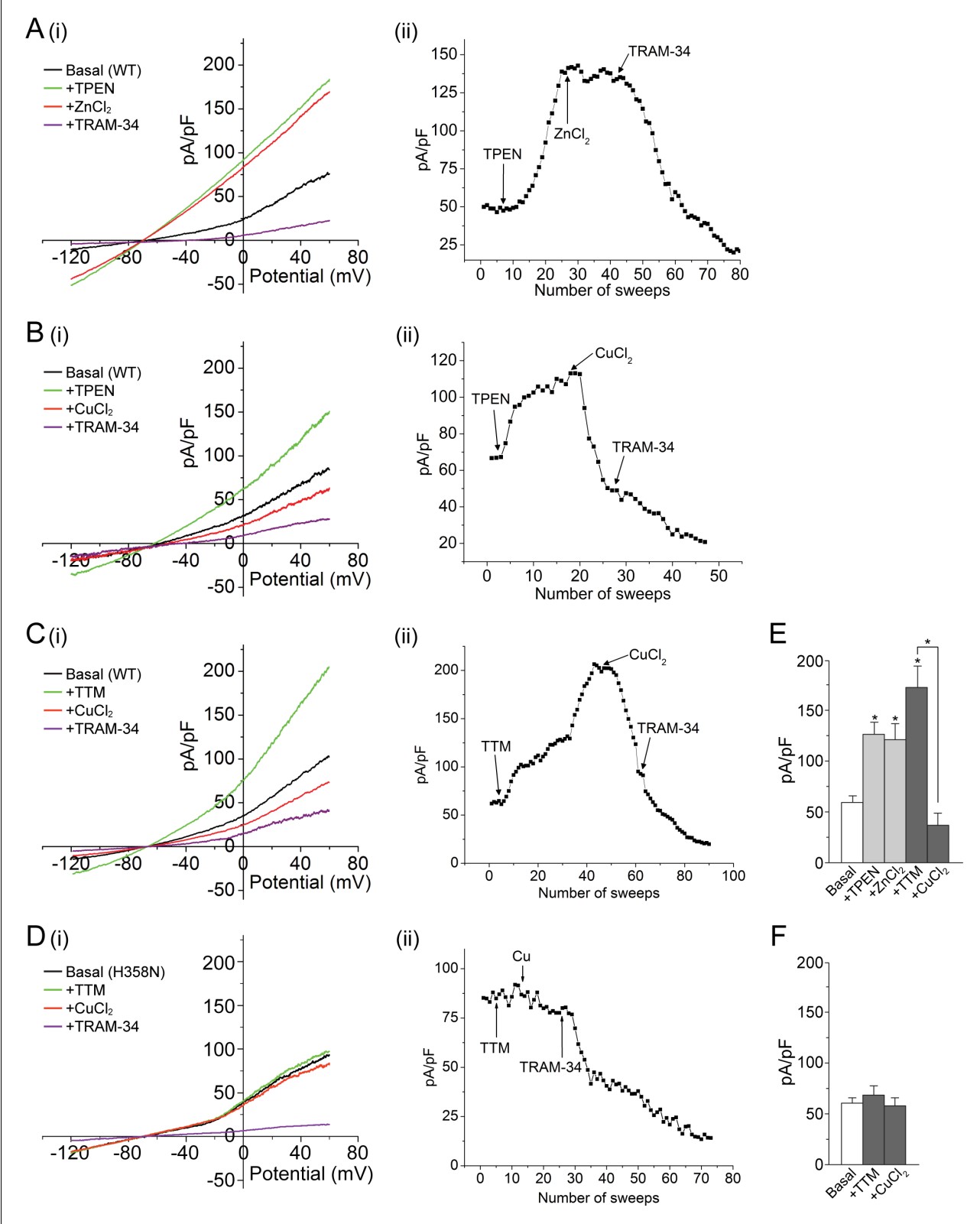

**Figure 1.** Metal chelators activate and copper inhibits KCa3.1 in whole-cell membrane patches. (**A–D**) (i) Representative current vs. voltage (IV) plot recorded from a 293-KCa3.1(WT) cell (**A–C**) or a 293-KCa3.1(H358N) cell (**D**). After obtaining the basal current, TPEN (20 µM) (**A**, **B**) or TTM (20 µM) (**C**, **D**) was perfused in the bath solution, followed by the addition of either ZnCl$_2$ (100 µM) (**A**) or CuCl$_2$ (100 µM) (**B–D**), and then TRAM-34 (1 µM). (**A–D**) (ii) Representative trace of current (pA/pF) recorded at +60 mV using ramp protocol applied every 10 s (sweep frequency). The timing of additions (TPEN,

*Figure 1 continued on next page*

*Figure 1 continued*

CuCl$_2$, etc.) is indicated by the arrows. (**E**, **F**) Summary bar graph of TRAM-34-sensitive current at +60 mV for data measured from 293-KCa3.1(WT) cells (**E**) or from 293-KCa3.1(H358N) cells (**F**). In (**E**), +ZnCl$_2$ is to be compared with +TPEN (i.e. ZnCl$_2$ added after treatment with TPEN), and +CuCl$_2$ with +TTM (i.e. CuCl$_2$ added after treatment with TTM). Data are displayed as mean ± the standard error of the mean (SEM) (n = 10 (**E**) or 8 (**F**) cells). *p≤0.01 versus Basal and for +CuCl$_2$ versus +TTM.

Because copper but not zinc inhibited KCa3.1 when added to cells, we also tested the copper-selective chelator tetrathiomolybdate (TTM) (*Brewer et al., 2006*). Similar to TPEN, addition of 20 µM TTM to 293-KCa3.1 cells caused a substantial increase in the measured current, which was reversed by the addition of 100 µM CuCl$_2$ (*Figure 1C,E*).

To determine whether the observed copper inhibition of KCa3.1 was mediated by His358, we transfected HEK 293 cells with the KCa3.1 mutant H358N and tested the effect of TTM and copper on channel activity. In contrast to wild-type (WT) KCa3.1, TPEN did not activate substantially, nor did copper inhibit substantially, KCa3.1 H358N (*Figure 1D,F*), demonstrating that copper inhibition of KCa3.1 is strictly dependent on His358.

## KCa3.1 is activated by TTM and inhibited by copper in isolated inside-out membrane patches, dependent on the phosphorylation state of His358

The direct effects of TTM and copper on KCa3.1 channel activity were further explored in inside-out (I/O) patch-clamp experiments using membrane patches pulled from 293-KCa3.1 cells. These experiments were designed to explore the model that KCa3.1 is inhibited by copper binding to His358, which is released by GTP/ATP-dependent NDPK-B phosphorylation of His358 or, in vitro, by chelation by TTM, with either inhibitory-release mechanism requiring calcium binding to calmodulin for channel activation. We first determined whether the addition of 20 µM TTM to the bath could activate KCa3.1, which it did (*Figure 2A*). We then added 10 µM copper (CuCl$_2$) to the bath, which decreased the current to the basal level. Re-application of TTM activated the channel (*Figure 2A*), demonstrating that inhibition and activation by copper and TTM, respectively, are reversible. NDPK-B also activated copper-inhibited KCa3.1, to a similar extent as TTM (*Figure 2B*). However, while NDPK-B required GTP to be present to activate KCa3.1 (*Figure 2C*), as GTP (or ATP) is necessary for NDPK-B to phosphorylate His358, TTM did not require GTP to activate KCa3.1 (*Figure 2A, B*), consistent with TTM simply chelating copper from His358.

Addition of 1 µM TRAM-34 suppressed channel activity to the basal level in all I/O patch experiments (*Figure 2A–D,F*), confirming that the measured activity is due to KCa3.1. The TTM-mediated current increase in the I/O patches was further confirmed to be KCa3.1-specific by demonstrating that addition of TTM after treatment with TRAM-34 did not increase the current (*Figure 2A*), as well as by the inability of either TTM or NDPK-B to elicit an increase in current in I/O patches isolated from parental (non-KCa3.1-transfected) 293 cells (*Figure 2G*).

To provide further evidence that phosphorylation by NDPK-B is the key event in regulating copper sensitivity of KCa3.1, we assessed the effects of TTM and copper on KCa3.1 activity after phosphorylation by NDPK-B. We found that copper did not inhibit KCa3.1 channel activity following prior activation by NDPK-B (*Figure 2C*), which would be predicted if phosphorylation of the imidazole ring of His358 abrogates copper binding. This is in contrast to the ability of copper to inhibit a TTM-activated channel (*Figure 2A,B*), in which case copper is free to bind to His358 once TTM is removed. Moreover, the addition of TTM did not lead to further activation of KCa3.1 (*Figure 2C*), as there should be little or no remaining His358-bound copper for TTM to chelate.

The experiments in *Figure 2A–C* were performed in the presence of EGTA to buffer calcium in the bath solution. In addition to chelating calcium, EGTA also chelates copper (*Xiao and Wedd, 2010*). While 10 µM CuCl$_2$ was required to inhibit KCa3.1 in an I/O patch in the presence of EGTA, 100 nM CuCl$_2$ was sufficient to inhibit in the absence of EGTA (*Figure 2D*). Furthermore, the ability of copper to inhibit KCa3.1 is inversely dependent on the calcium concentration: approximately 1 µM CuCl$_2$ was required for inhibition at 10 µM CaCl$_2$ (*Figure 2E*), versus 100 nM CuCl$_2$ at 300 nM CaCl$_2$ (*Figure 2D*). The copper (*Figure 2E*) and TRAM-34-inhibited currents (*Figure 2F*) in 10 µM

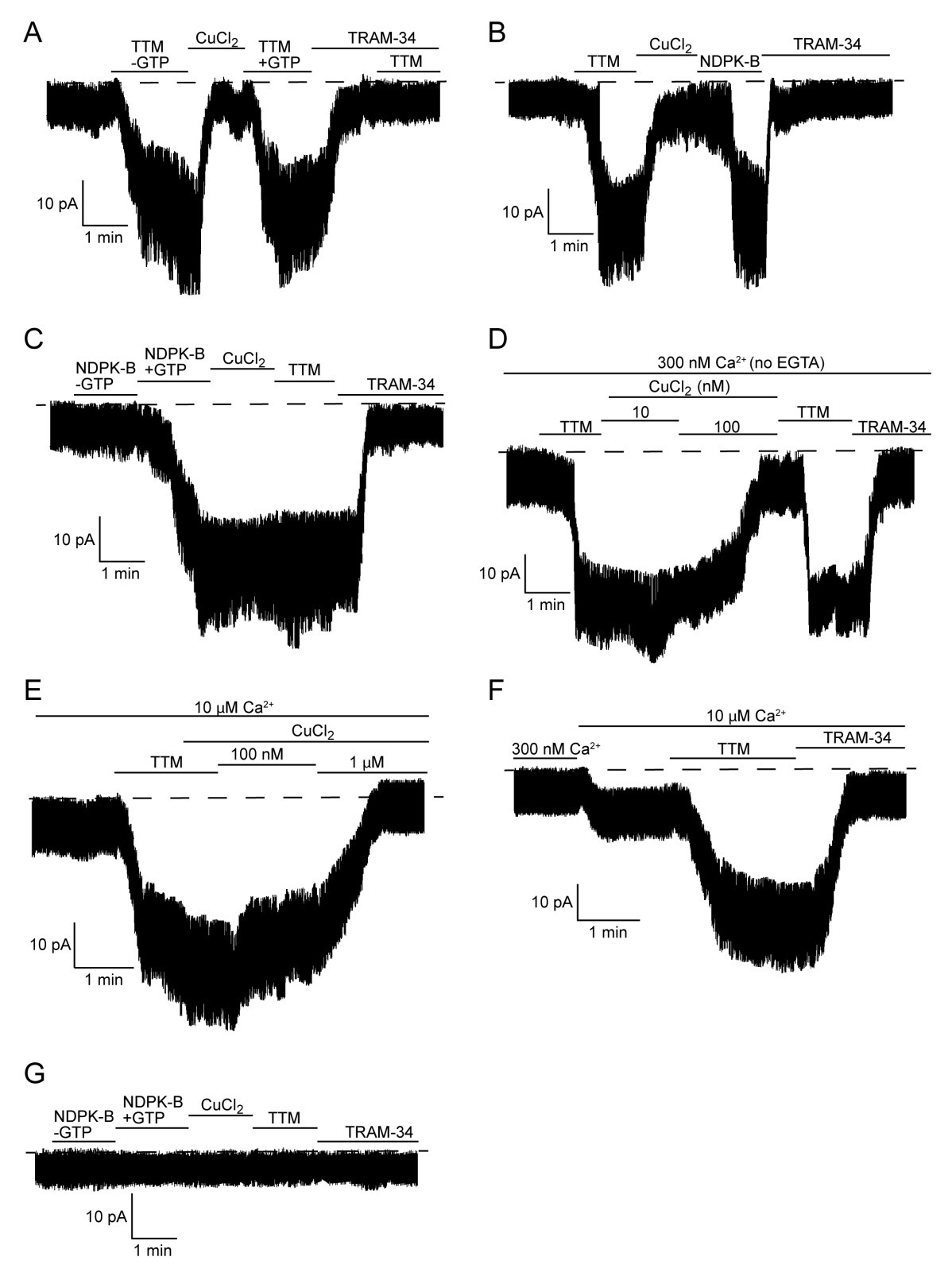

**Figure 2.** TTM activates and copper inhibits KCa3.1 in inside-out membrane patches, dependent on His358 phosphorylation. Representative recordings of channel activity versus time from I/O patches isolated from 293-KCa3.1 (**A–F**) or parental 293 (**G**) cells. All recordings were done in 300 nM free $Ca^{2+}$ and 500 µM GTP unless otherwise specified. TTM (20 µM), $CuCl_2$ (10 µM), NDPK-B (10 µg/ml) and TRAM-34 (1 µM) were added as indicated. Shown are

*Figure 2 continued on next page*

*Figure 2 continued*

representative patch data from three independent experiments. (**A–C**) were performed in the presence of EGTA (5 mM) and (**D–F**) in the absence of EGTA in the bath solution. (**E,F**) were done in the presence of 10 µM free calcium in the bath solution.

calcium, following the addition of TTM, were the same, indicating that the copper-inhibited current is specific to KCa3.1.

We showed previously that activation of KCa3.1 requires both His358 phosphorylation by NDPK-B and calcium binding to KCa3.1-bound calmodulin (*Srivastava et al., 2006b*). We therefore tested whether calcium is also required for activation by TTM. Addition of 20 µM TTM to an I/O patch in low calcium (30 nM) did not result in KCa3.1 activation (*Figure 3A*), indicating that calcium is also required for TTM activation of KCa3.1.

If copper bound to His358 is chelated by TTM in low calcium, we anticipated that perfusing the patch with high calcium (300 nM) following TTM would be sufficient to activate. Surprisingly, addition of 300 nM calcium to an I/O patch following 20 µM TTM did not activate KCa3.1, although it did activate other non-KCa3.1 channels that were not inhibited by TRAM-34 (*Figure 3A*). Rather, TTM and high calcium needed to be present simultaneously for activation (*Figure 3A*). A similar result was obtained for activation of KCa3.1 by NDPK-B (*Figure 3B*). These data suggest that a calcium/calmodulin-induced conformational change in the channel is required for TTM and NDPK-B to gain access to copper and His358, respectively. Addition of copper inhibited the TRAM-34-sensitive current, but not the current from other channels activated by calcium (*Figure 3A*), providing further evidence that copper specifically inhibits KCa3.1.

## Basal KCa3.1 current is elevated in transfected MEFs from *Ctr1*$^{-/-}$ mice

To further explore our finding that copper inhibits KCa3.1 activity, we measured current from GFP-KCa3.1-transfected MEFs derived from mice in which the gene for the copper transporter Ctr1 was deleted (*Lee et al., 2002*). As shown in *Figure 4A–C*, the basal current is approximately three-fold higher in the KCa3.1-transfected *Ctr1*$^{-/-}$ MEFs than in the transfected WT MEFs expressing similar amounts of GFP-KCa3.1 (only GFP$^{+}$ cells exhibiting similar levels of immunofluorescence were patched). Furthermore, in contrast to the transfected WT MEFs (*Figure 4A,C*), addition of 20 µM TTM had little effect on the current from the transfected *Ctr1*$^{-/-}$ MEFs (*Figure 4B,C*). Addition of

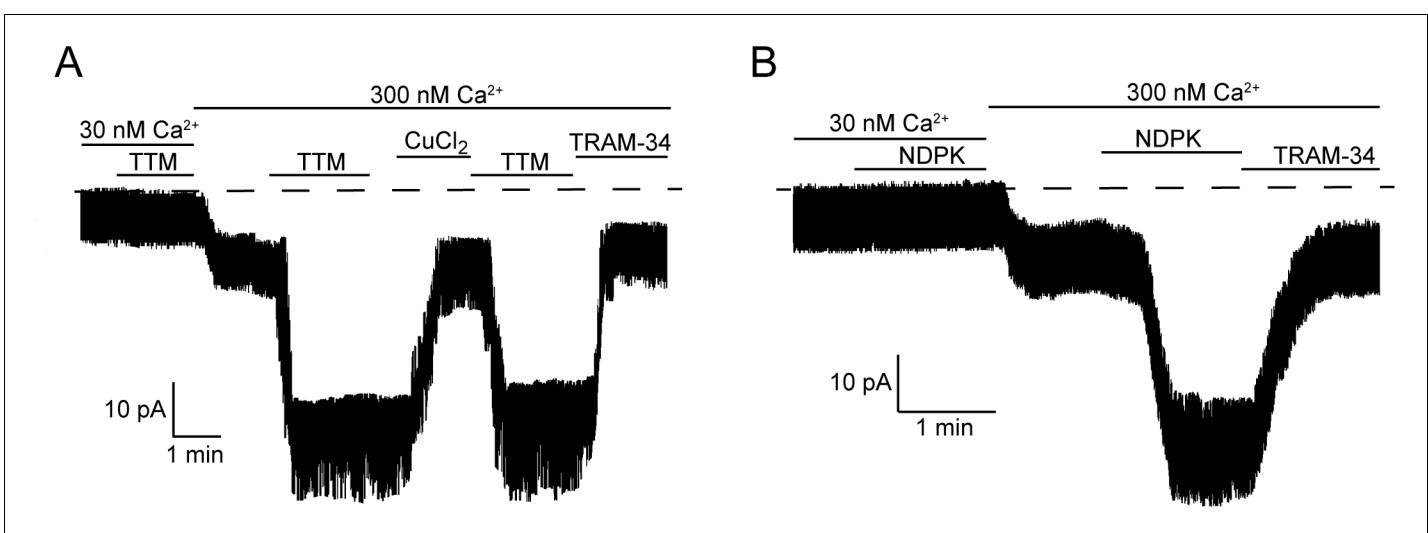

**Figure 3.** Calcium is required for activation of KCa3.1 by NDPK-B or TTM in inside-out membrane patches. Representative recordings of channel activity versus time from I/O patches isolated from 293-KCa3.1 cells as described in *Figure 2*. All recordings were done in 500 µM GTP and either 30 or 300 nM free Ca$^{2+}$ as indicated. TTM (20 µM), NDPK-B (10 µg/ml), CuCl$_2$ (10 µM) and TRAM-34 (1 µM) were added as indicated. Shown are representative patch data from three independent experiments.

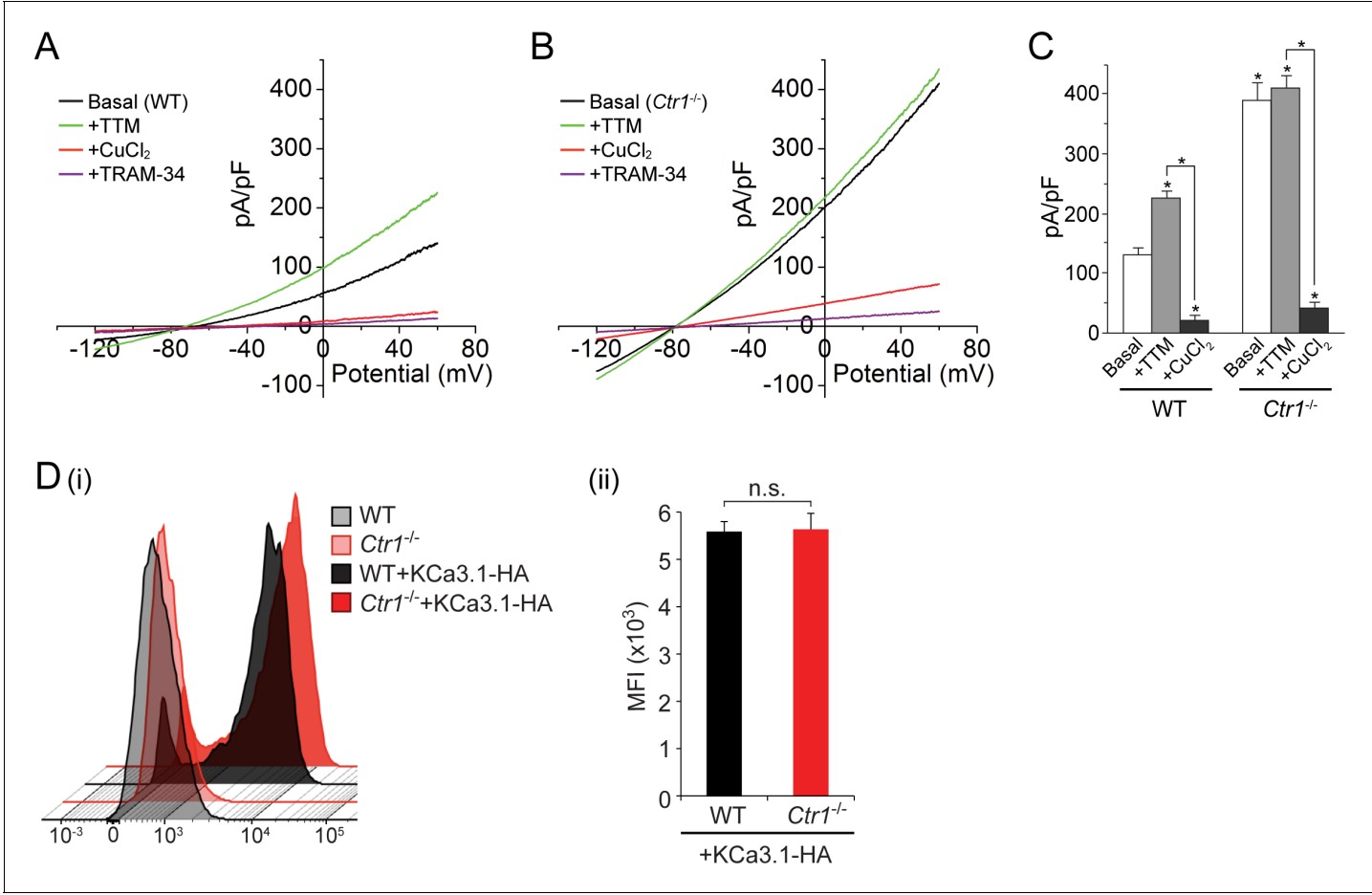

**Figure 4.** Basal KCa3.1 current is elevated in transfected MEFs from $Ctr1^{-/-}$ mice. (A, B) Representative current vs. voltage (IV) plot recorded from a KCa3.1-transfected MEF from a wild-type (WT) mouse (A) or a $Ctr1^{-/-}$ (B) mouse. After obtaining the basal current, TTM (20 μM) was perfused in the bath solution followed by $CuCl_2$ (10 μM) and then TRAM-34 (1 μM). (C) Summary bar graph of the TRAM-34-sensitive current at +60 mV for data measured from MEFs from WT or $Ctr1^{-/-}$ mice. For statistical analysis, one-way-ANOVA was used, and the Bonferroni test was applied to compare the mean values. Data are displayed as mean ± SEM (n = 10 cells). *p≤0.01 versus Basal in WT MEFs and for +$CuCl_2$ versus +TTM. (D) Exofacial HA-tagged KCa3.1 was transfected into WT or $Ctr1^{-/-}$ MEFs, and cell surface expression was assessed by FACs analysis following staining with anti-HA antibodies and with anti-mouse FITC antibodies in non-permeabilized cells. (i) Flow cytometry results of WT and $Ctr1^{-/-}$ controls stained with only the secondary anti-mouse FITC antibody, or of WT + KCa3.1-HA and $Ctr1^{-/-}$ + KCa3.1-HA MEFs stained with both anti-HA and anti-mouse FITC antibodies. (ii) Mean fluorescence intensity (MFI) of WT + KCa3.1-HA and $Ctr1^{-/-}$ + KCa3.1-HA MEFs.

exogenous copper (10 μM) inhibited KCa3.1 current in the transfected $Ctr1^{-/-}$ MEFs as well as the WT MEFs (*Figure 4A–C*). Entry of copper into $Ctr1^{-/-}$ MEFs was presumably mediated by a Ctr1-independent process, as has been described (*Lee et al., 2002*). The failure of exogenous TTM to increase KCa3.1 current in WT MEFs to levels similar to those in $Ctr1^{-/-}$ MEFs (*Figure 4A,B*) is likely due to incomplete chelation of copper bound to His358.

To rule out the possibility that the increased KCa3.1 current in the $Ctr1^{-/-}$ MEFs was due to an increase in surface expression of KCa3.1 in these MEFs, we transfected WT and $Ctr1^{-/-}$ MEFs with KCa3.1 bearing an exofacial hemagglutinin (HA) epitope tag inserted into the extracellular loop between transmembrane helices S3 and S4 (*Srivastava et al., 2005*). Previous studies (*Srivastava et al., 2005*; *Syme et al., 2003*) have demonstrated that KCa3.1 with this tag functions normally and can be specifically detected at the plasma membrane following staining of non-permeabilized cells with anti-HA antibody. FACS analysis of non-permeabilized cells demonstrated that the surface levels of KCa3.1 in the two types of MEFs were very similar (*Figure 4D*).

# CD4$^+$ T cells derived from *Ctr1$^{+/-}$* mice are hyperactivated and endogenous KCa3.1 is hyperphosphorylated on histidine and exhibits increased activity

Finally, we tested whether copper inhibition of KCa3.1 is physiologically relevant in the context of CD4$^+$ T-cell activation. By effluxing potassium ions, KCa3.1 maintains a negative membrane potential required for additional calcium influx. Because *Ctr1$^{-/-}$* mice are embryonically lethal, we assessed T-cell activation in CD4$^+$ T cells from mice that were heterozygous for *Ctr1*. The level of cytoplasmic copper in the *Ctr1$^{+/-}$* splenocytes is approximately 50% of that in WT splenocytes (*Lee et al., 2001*). As shown in *Figure 5A*, endogenous KCa3.1 channel activity was markedly increased in CD4$^+$ T cells isolated from *Ctr1$^{+/-}$* mice compared to those from WT littermate controls. Consistent with increased KCa3.1 activity, by blotting with a monoclonal antibody specific for 3-pHis (*Fuhs et al., 2015*), we observed significantly increased histidine phosphorylation of (endogenous) KCa3.1 in T cells from *Ctr1$^{+/-}$* mice versus those from WT littermate controls, with no appreciable difference in KCa3.1 expression levels (*Figure 5B*). In accord with elevated KCa3.1 activity, T-cell receptor-stimulated (by anti-CD3/CD28 antibodies) calcium influx and cytokine production—interleukin-2 (IL-2), interferon-γ (IFN-γ) and tumor necrosis factor-α (TNFα)—were also substantially increased in *Ctr1$^{+/-}$* T cells (*Figure 5C–E*).

## Discussion

KCa3.1 is unique among the four calcium-activated KCa potassium channels in that activation requires phosphorylation by NDPK-B of a specific histidine residue, His358, in the cytoplasmic domain (*Srivastava et al., 2006b*). The mechanism by which (non-phosphorylated) His358 is inhibitory to KCa3.1 was not known. The specificity of His358 inhibition—the conservative substitution asparagine completely disinhibits the channel—pointed to an inhibitory mechanism for which histidine is uniquely suited.

One important biochemical feature of histidine is its role in coordinating metal ions, through one of the two nitrogen atoms (N1 and N3) in the imidazole ring. Metal ions for which histidine serves as a ligand include copper, nickel, iron, zinc and manganese (*Zheng et al., 2008*). In homotetrameric KCa3.1, four copies of His358 are present in the cytoplasmic region, related (on average) by C4 symmetry. Of the metal ions listed above, only copper (Cu(II)) is found to be coordinated solely by four histidine residues (*Dokmanić et al., 2008*).

These considerations suggested the possibility that the KCa3.1 inhibitory mechanism, mediated by His358, involves copper binding. In the present study, we provide strong support for this hypothesis. Moreover, we show that copper inhibition of KCa3.1 is physiologically relevant for CD4$^+$ T-cell activation; KCa3.1 in T cells deficient in intracellular copper (from *Ctr1$^{+/-}$* mice) exhibits higher activity, which results in increased calcium influx and cytokine production (*Figure 5*). Although the BK and Shaker potassium channels and TASK-3, a two-pore-domain potassium channel, have also been shown to be inhibited by copper (*Gruss et al., 2004*; *Ma et al., 2008*), the inhibition mechanism is distinct from that in KCa3.1 because copper acts extracellularly in these other channels, and there is no evidence that these effects are regulated by histidine phosphorylation.

How would His358 be positioned spatially to bind a copper ion? His358 is located just C-terminal to the calmodulin-binding domain in KCa3.1 and 18 residues N-terminal to the predicted C-terminal coiled-coil region, which crystallographic analysis shows forms a four-helix bundle (S.R.H., unpublished data). Coiled-coil prediction software indicates that there is a reasonable probability that the region encompassing His358 also forms a coiled-coil/four-helix bundle, with His358 occupying an inward-facing 'a' position in the heptad repeat, which would position the four copies of His358 to coordinate a copper ion (via N3 in the imidazole ring) on the four-fold axis. Biophysical studies are ongoing to validate this structural model and demonstrate direct binding of copper to His358.

Our model for the phosphorylation- and calcium-dependent activation of KCa3.1 is presented in *Figure 6*. In the absence of calcium (basal state), copper binding to the four copies of His358 in homotetrameric KCa3.1 maintains low channel activity by opposing the calcium/calmodulin-mediated conformational changes that induce channel opening. Upon a marked increase in intracellular calcium levels due to T-cell stimulation (for example), calcium binding to channel-associated calmodulin induces conformational changes in the channel, distorting the four-helix bundle locally (the C-terminal four-helix bundle is maintained) and partially destabilizing copper binding to His358, thus

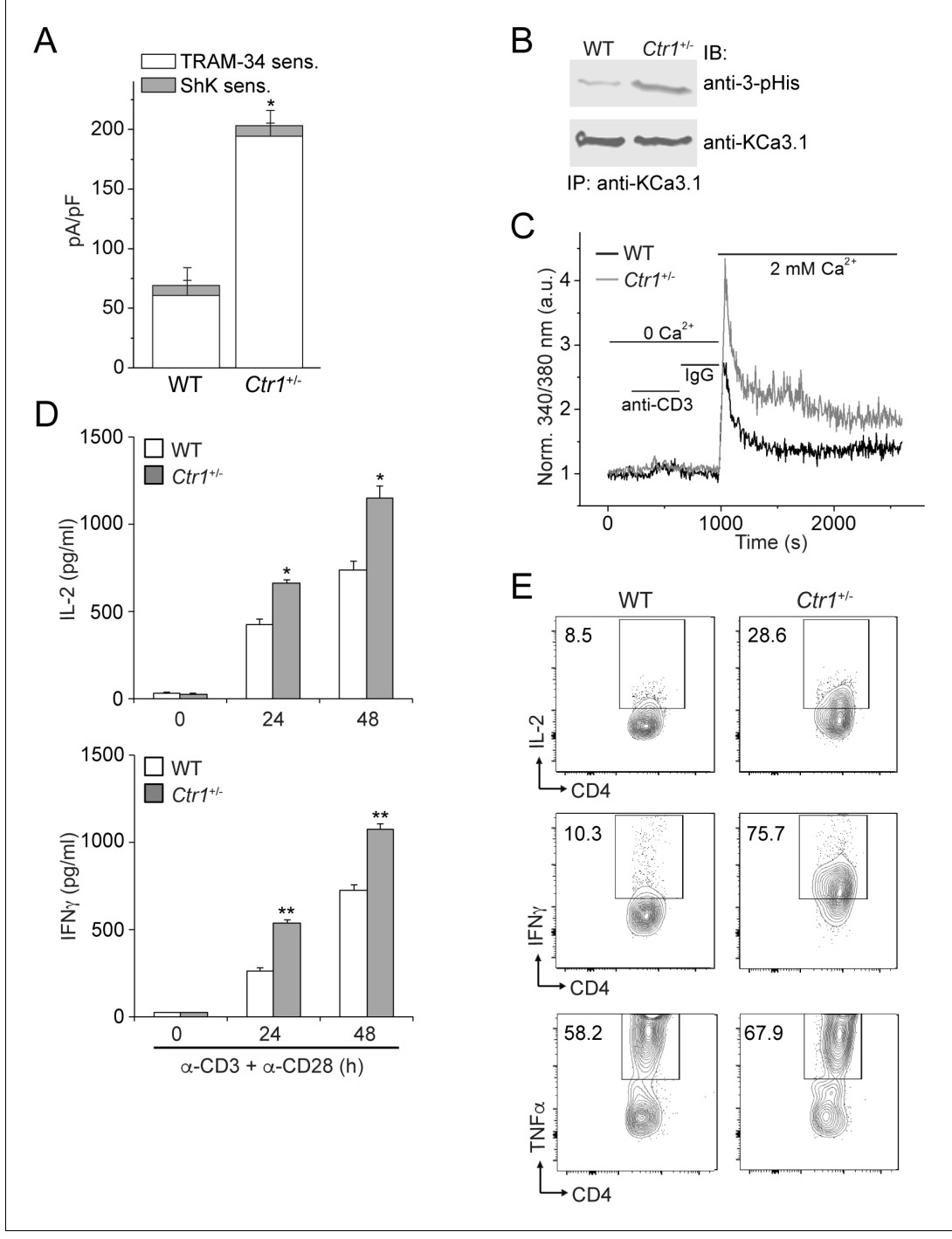

**Figure 5.** CD4[+] T cells from *Ctr1[+/-]* mice are hyperactivated. Naive CD4[+] T cells were isolated from spleens of wild-type (WT) or *Ctr1[+/-]* mice and activated with anti-CD3 and -CD28 antibodies for 48 hr. (**A**) KCa3.1 channel activity was determined by whole-cell patch clamping as in *Figure 1*. Shown is a summary bar graph of TRAM-34 (1 μM; KCa3.1) and ShK (1 nM; Kv1.3)-sensitive currents at +60 mV (n = 12 (WT) or 15 (*Ctr1[+/-]*)). For statistical analysis, one-way-ANOVA was used, and the Bonferroni test was applied to compare the mean values. Data are displayed as mean ± SEM (n = 10 cells). *p≤0.01 vs TRAM-34-sensitive current in WT. (**B**) KCa3.1 was immunoprecipitated from lysates of CD4[+] T cells from WT or *Ctr1[+/-]* mice and probed with a monoclonal antibody to 3-pHis (clone SC56-2) or KCa3.1 as indicated. (**C**) Activated cells were rested overnight, loaded with Fura-2, AM and attached to a poly-L-lysine-coated coverslip for 20 min. Calcium imaging was then performed in unstimulated cells and following stimulation with anti-CD3 and -CD28 antibodies (n = 80–100 cells). (**D**) ELISA to quantify IL-2 and IFN-γ in the supernatants. Statistical significance was calculated using Student's *t*-test (*p<0.05, **p<0.01, n.s.: *Figure 5 continued on next page*

*Figure 5 continued*

p>0.05; not significant as compared to WT). (E) Representative intracellular flow cytometry detecting cytokine expression following restimulation with anti-CD3 and -CD28 antibodies in the presence of brefeldin-A for 4 hr. Cells not re-stimulated with anti-CD3 and -CD28 were stained and analyzed in a similar manner and served as controls for setting the cut-off limits for cytokine production. Data are representative of two independent experiments.

exposing it to phosphorylation by NDPK-B. Because stable copper binding requires tetravalent coordination, phosphorylation of a single His358 residue (on N3) in the tetramer is probably sufficient to abrogate copper binding and inhibition of KCa3.1. Unencumbered by copper, the channel-opening conformational changes induced by calcium proceed in the same manner in KCa3.1 as in the other three KCa channels. Consistent with our model, (i) NDPK-B or TTM activation of KCa3.1 required calcium to be present concurrently (*Figure 3*), (ii) copper did not inhibit NDPK-B-activated (phosphorylated) KCa3.1 (*Figure 2C*), and (iii) KCa3.1 was hyperphosphorylated on histidine (and more active) in CD4$^+$ T cells from *Ctr1*$^{+/-}$ mice (*Figure 5B*), which contain lower intracellular copper levels.

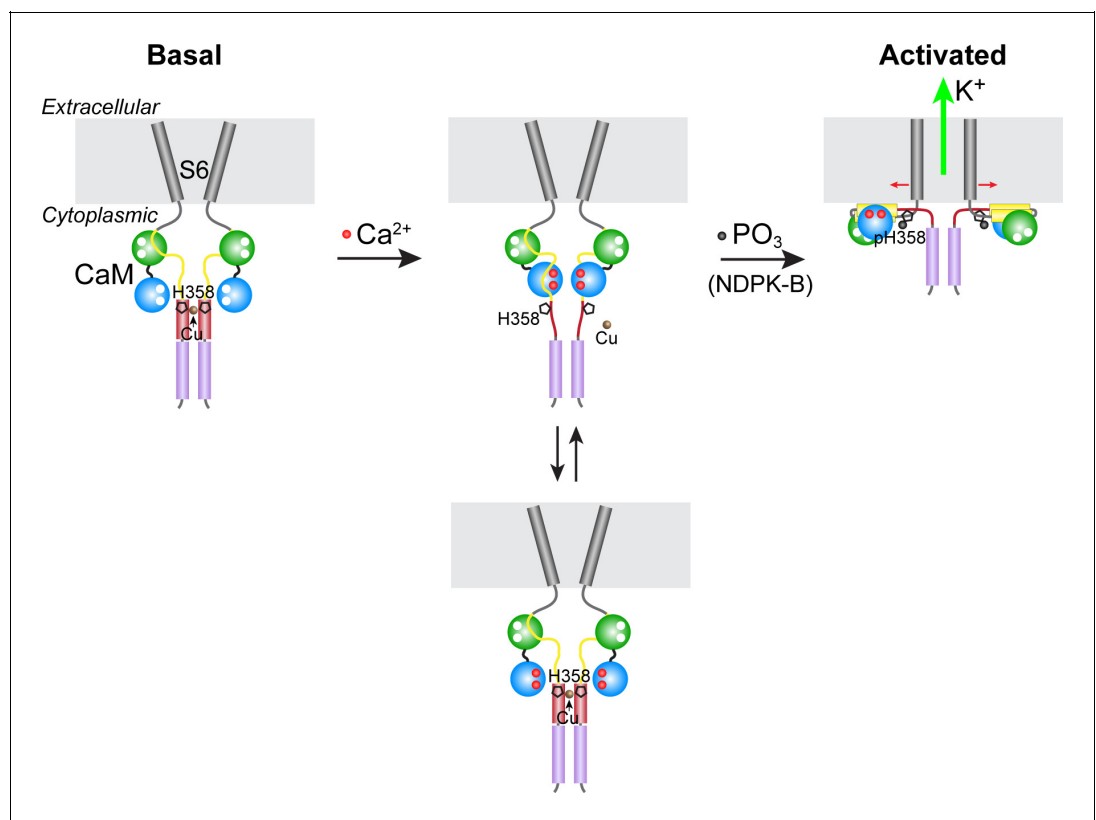

**Figure 6.** Model for KCa3.1 activation by calcium and histidine phosphorylation. For clarity, only two of the four KCa3.1 subunits and calmodulin (CaM) are shown. In the basal state (no calcium; left panel), the CaM C lobe (green sphere) is bound to the N-terminal segment of the calmodulin-binding domain (CBD, yellow) in KCa3.1 (*Schumacher et al., 2004*). The transmembrane S6 helices close off the channel on the cytoplasmic side. At the C terminus of KCa3.1 is a coiled-coil region that forms a four-helix bundle (violet cylinders; S.R.H., unpublished data). Based on coiled-coil prediction software, it is probable that the region containing His358, which is just C-terminal to the CBD, also forms a four-helix bundle (maroon cylinders), with His358 (pentagon) occupying an inward-facing 'a' position in the heptad repeat. This would position the four copies of His358 for coordination of a Cu (II) ion on the axis of the four-helix bundle, which would stabilize the four-helix bundle and act to resist the conformational changes induced by calcium binding to the CaM N lobe (blue sphere). An increase in intracellular calcium induces conformational changes in the CBD that partially destabilize copper binding (middle panel), providing access to His358 for phosphorylation by NDPK-B (or copper chelation by TTM). Upon phosphorylation of His358, copper binding is abrogated, and the calcium-induced conformational changes in the CBD lead to channel opening (right panel; exact mechanism not known) (*Adelman et al., 2012*; *Sachyani et al., 2014*).

The metal-binding properties of histidine and its ability to be phosphorylated distinguish histidine from the common phosphorylatable residues in eukaryotes, serine/threonine and tyrosine. Although our model for KCa3.1 inhibition and activation might be unique for a potassium channel, we recently demonstrated that the TRPV5 calcium channel is also activated by NDPK-B-mediated histidine phosphorylation (*Cai et al., 2014*). Whether metal-ion coordination by histidine, disrupted through histidine phosphorylation, is the basis for regulation of TRPV5, or of any of the 1- and 3-pHis-containing proteins recently identified with anti-pHis antibodies (*Fuhs et al., 2015*), remains to be explored.

Our finding that limiting intracellular copper can enhance activation of T cells has therapeutic implications. It was recently shown that copper chelators such as penicillamine, used to treat Wilson disease (copper-overload disorder), were able to slow growth of oncogenic BRAF$^{V600E}$-driven tumors in mice, through inactivation of MEK1 (*Brady et al., 2014*). In addition to directly targeting BRAF$^{V600E}$ in tumors, treatment with checkpoint inhibitors to enhance immune-cell killing has led to dramatic response rates (*Pardoll, 2012*). Thus, treatment of patients harboring BRAF$^{V600E}$ driven-tumors with copper chelators has the potential to not only block BRAF activation of the MAP kinase pathway, but also to enhance T-cell-mediated killing of these tumors via enhanced activation of KCa3.1.

# Materials and methods

## Materials

TPEN (Sigma-Aldrich, St. Louis, MO, Cat # P4413), TTM (Sigma-Aldrich, Cat # 323446), CuCl$_2$ (Sigma-Aldrich, Cat # C6641) and ZnCl$_2$ (Acros, Geel, Belgium, Cat # 318170100) were purchased. TRAM-34 was obtained from Heike Wulff (University of California–Davis). $Ctr1^{-/-}$ MEFs and $Ctr1^{+/-}$ splenocytes have been described previously (*Lee et al., 2001*, *2002*).

## Whole-cell patch clamping

HEK 293 cells stably transfected with GFP-tagged human KCa3.1 (293-KCa3.1) (in vector pEGFP-C1), either WT or H358N, or MEFs from WT or $Ctr1^{-/-}$ mice transiently transfected with mCherry-tagged KCa3.1 were used for electrophysiological studies. Cells were plated on 15-mm diameter coverslips and, after 24 hr, patch clamping was performed at room temperature as described (*Srivastava et al., 2005*), using a pipette solution containing 100 mM K-gluconate, 30 mM KCl, 1.2 mM MgCl$_2$, 5 mM EGTA, 4.3 mM CaCl$_2$ and 10 mM HEPES, pH 7.2, with 1 N KOH (calculated free Ca$^{2+}$, 1 μM), and a bath solution containing 140 mM NaCl, 5 mM KCl, 1 mM CaCl$_2$, 1 mM MgCl$_2$, 10 mM glucose and 10 mM HEPES, pH 7.4. Patch-clamp pipettes had resistances ranging between 3 and 3.5 MΩ. Current-voltage (I-V) relationships were measured using ramp voltage-clamp protocols (at 15-s intervals) from a holding potential of −70 to −120 mV, followed by ramp depolarization to +60 mV (symmetrical ramp rate of 0.18 mV·ms$^{-1}$). The current-voltage relationship was obtained by plotting the current during the depolarizing ramp phase as a function of the corresponding voltage. Membrane currents were filtered (−3 dB at 1 kHz) and digitized at 10 kHz (pClamp 10.3 with Digidata 1440A ADC interface; Molecular Devices, Sunnyvale, CA). Cell capacitance and pipette series resistances were compensated (usually > 80%). Whole-cell current density was expressed as picoamperes per picofarad (pA/pF).

## Inside-out (I/O) patch clamping

293-KCa3.1 cells were used for single-channel recordings, performed using the I/O patch-clamp mode. Data were obtained using standard patch-clamp techniques with an Axopatch-200B amplifier (Molecular Devices). Currents were recorded at room temperature (21–23°C), low-pass filtered (-3-dB cut-off frequency at 1 kHz), and recorded on computer disk at a sampling frequency of 5 kHz (Clampex; Molecular Devices). The pipette resistance was 3–4 MΩ when filled with a solution containing 145 mM KCl, 1 mM MgCl$_2$, 1 mM CaCl$_2$ and 5 mM HEPES, pH 7.4. The bath solution contained 141 mM KCl, 5 mM EGTA, 1 mM MgCl$_2$, 3.3 mM CaCl$_2$, 10 mM glucose and 5 mM HEPES, pH 7.2. The calcium activity was calculated to be 300 nM (Win-MAX Chelator software) (*Bers et al., 1994*). Data were analyzed using pClamp 10.3 (Molecular Devices) and Origin 7.0 (OriginLab, Northampton, MA) software. GST-NDPK-B was expressed in *E. coli* (pGEX-4T-2, GE Healthcare, Chicago, IL) and purified by Glutathione-Sepharose chromatography (GE Healthcare).

## Characterization of CD4$^+$ T cells isolated from WT and *Ctr1$^{+/-}$* mice

Mouse CD4$^+$ T cells were isolated from spleens of WT and *Ctr1$^{+/-}$* mice using the mouse CD4$^+$ T-cell isolation kit from Miltenyi Biotec (Bergisch Gladbach, Germany) according to the manufacturer's protocol. We routinely obtained > 95% CD4$^+$ T cells as assessed by FACS. Whole-cell patch clamping was then performed following stimulation with anti-CD3 (5 µg/ml) and anti-CD28 (2 µg/ml) antibodies for 48 hr, as described (*Srivastava et al., 2008*). For intracellular cytokine staining, cells were rested overnight and then re-stimulated with anti-CD3 and -CD28 antibodies for 4 hr in the presence of brefeldin-A (1 µg/ml, Sigma-Aldrich). Intracellular staining was performed using the Fixation/Permeabilization kit (BD Biosciences, Franklin Lakes, NJ, Cat # 554714) and antibodies to: IL-2 (Cat # JES6-5H4), IFN-γ (Cat # XMG1.2) or TNFα (Cat # MP6-XT22), according to the manufacturer's (eBioscience, San Diego, CA) instructions. Cells were analyzed on a LSRII HTS flow cytometer (BD Biosciences), and the results were analyzed with Treestar (FlowJo, Ashland, OR).

## Intracellular calcium flux

Mouse CD4$^+$ T cells were loaded with 5 µm Fura-2, AM (Molecular Probes, Eugene, OR) in RPMI 1640 medium and 10% FBS for 30 min at 22–25°C. Cells were then attached to a poly-L-lysine-coated coverslip for 20 min, and calcium imaging was done with an IX81 epifluorescence microscope (Olympus, Tokyo, Japan) and OpenLab imaging software (Improvision, Conventry, UK) as described (*Di et al., 2010*). For single-cell analysis, 340/380 nm Fura-2 emission ratios of > 100 cells per experiment were analyzed. Background fluorescence obtained from regions containing no cells was digitally subtracted from each image. To compare the different groups, the 340/380 nm ratio was normalized to 1 by dividing the fluorescence values at different time points to the cellular fluorescence at time 0. Experiments were independently repeated at least three times.

## Immunoprecipitation and western blotting

Mouse CD4$^+$ T cells were lysed in lysis buffer (100 mM NaCl, 10% glycerol, 1 mM EDTA, 5 mM MgCl$_2$, 0.5% NP-40 and 50 mM Tris, pH 8.0, along with freshly added 1 mM DTT, protease and phosphatase inhibitors) and immunoprecipitated with anti-KCa3.1 antibodies (Alomone Labs, Jerusalem, Israel, Cat # ALM-051). After washing three times with lysis buffer, bound proteins were eluted using SDS loading buffer (pH 8.8), separated by SDS-PAGE (pH 8.8), and immunoblotted with anti-KCa3.1 or rabbit monoclonal anti-3-pHis (clone SC56-2) antibodies (*Fuhs et al., 2015*).

## FACS analysis of MEFs transfected with exofacial HA-tagged KCa3.1

KCa3.1 containing a hemagglutinin (HA) epitope tag inserted into the extracellular loops between transmembrane helices S3 and S4 was transfected into *Ctr1$^{-/-}$* and WT MEFs. To assess the plasma membrane levels of KCa3.1, cells were fixed with 4% paraformaldehyde for 10 min on ice and washed three times with PBS. The non-permeabilized fixed cells were incubated with anti-HA antibody (1:200 dilution) for 1 hr on ice, washed three times with PBS, and stained with secondary anti-mouse FITC-conjugated antibody (1:500 dilution) for 30 min on ice. Cells were then washed three times with PBS prior to analysis by flow cytometry. Cells incubated with only the secondary antibody served as the negative control. Flow cytometry was performed using LSR II HTS flow cytometer (BD Biosciences), and the results were analyzed using Treestar (FlowJo).

## Acknowledgements

We thank Senena Corbalán-García (University of Murcia) for helpful discussions, Min Ha Hwang (Duke University) for mice dissections, and Heike Wulff (University of California–Davis) for TRAM-34. This work was supported in part by NIH grants R01CA194584 (TH), R01DK074192 (DJT), R21AI107443 (SRH), and R01GM084195 (EYS). TH is a Frank and Else Schilling American Cancer Society Professor and holds the Renato Dulbecco Chair in Cancer Research.

# Additional information

### Competing interests

TH: Senior editor, *eLife*. The other authors declare that no competing interests exist.

### Funding

| Funder | Grant reference number | Author |
|---|---|---|
| National Institute of Allergy and Infectious Diseases | R21AI107443 | Stevan R Hubbard |
| National Institute of Diabetes and Digestive and Kidney Diseases | R01DK074192 | Dennis J Thiele |
| National Institute of General Medical Sciences | R01GM084195 | Edward Y Skolnik |
| National Cancer Institute | R01CA194584 | Tony Hunter |

The funders had no role in study design, data collection and interpretation, or the decision to submit the work for publication.

### Author contributions

SS, SP, Acquisition of data, Analysis and interpretation of data, Drafting or revising the article; ZL, Acquisition of data, Contributed unpublished essential data or reagents; SRF, TH, DJT, Drafting or revising the article, Contributed unpublished essential data or reagents; SRH, EYS, Conception and design, Analysis and interpretation of data, Drafting or revising the article

### Author ORCIDs

Tony Hunter, http://orcid.org/0000-0002-7691-6993
Stevan R Hubbard, http://orcid.org/0000-0002-2707-9383

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
