## [Decision Letter]

Thank you for submitting your article "Histidine phosphorylation relieves copper inhibition in the mammalian potassium channel KCa3.1" for consideration by *eLife*. Your article has been reviewed by three peer reviewers, one of whom is a member of our Board of Reviewing Editors, and the evaluation has been overseen by Ivan Dikic as the Senior Editor.

The reviewers have discussed the reviews with one another and the Reviewing Editor has drafted this decision to help you prepare a revised submission.

Summary:

This study that establishes a potential mechanism for the regulation of KCa3.1 channels in CD4 T cells by histidine phosphorylation on His358. The authors propose that Cu^++^ ions would coordinate with four His358 residues thus rendering the channel refractory to the conformational changes initiated by the binding of Ca^2+^ to CaM. This mechanism is blocked by His358 phosphorylation. This may have important physiological significance in the context of understanding CD4 T cell activation.

Essential revisions:

1) Copper binding to His358 is inferred from indirect studies, but is not directly demonstrated – this should be acknowledged. The dose response of copper (Figure 1) shows that 1µM had no effect, 10µM shows an inhibitory effect, and that 100µM was saturating. How does this dose response compare with physiological intracellular copper concentrations?

2) The concentration of free Cu^++^ in the solutions used for the whole cell and inside-out patch clamp experiments is a concern. The expected Cu^++^ concentration in the pipette solution used in whole cell experiments and in the bath solution for the inside-out experiments should be specified. EDTA has a binding affinity of 10-18M for Cu^++^ and it is likely to be the same for EGTA. Under such conditions, both solutions should be Cu^++^ free and yet channel activity appears to be low in the whole cell and inside-out recordings presented in Figure 1 and Figure 2 in control conditions.

3) Figure 2 shows that channel activity remains high after TTM removal in 300nM Ca^2+^ (the effect of TTM is not reversible) consistent with the bathing solution being Cu^++^ free and TTM being capable of chelating Cu^++^ from the Cu^++^- H358 complex. However, channel activity appears low in 300nm Ca^2+^ after TTM treatment in 30nM Ca^2+^. One would have expected channel activity to be high in this case if TTM had succeeded to chelate Cu^++^ from the Cu^++^-H358 complex in 30nM Ca^2+^ conditions. The authors need to discuss how the action of TTM might be modulated by Ca^2+^ and may involve more than chelating Cu^++^.

4) Segments of the inside-out recording in Figure 2 should also be presented at a higher time resolution to compare single channel activity at 300nm Ca^2+^ + low Cu^++^, 10μM CuCl_2_ and TTM conditions. The authors also need to comment on the finding that in Figure 2, channel activity appears greater in CuCl_2_ conditions than in control before TTM application in 300nm Ca^2+^ conditions.

5) The proposed mechanism implies that Cu^++^ coordination by four H358 is state dependent, with Cu^++^ binding taking place in the channel-closed configuration. Accordingly, the potency of CuCl_2_ to decrease channel activity following TTM treatment should be Ca^2+^ dependent with CuCl_2_ being less potent in saturating Ca^2+^ conditions (10μM Ca^2+^). The Ca^2+^ dependency of CuCl_2_ mediated inhibition after TTM treatment should be presented in inside-out experiments to confirm state dependency.

6) The significance of the results illustrated in Figure 3 would be clearer if information on the mechanisms underlying Cu^++^ cellular homeostasis, and on the expected basal Cu^++^ intracellular level is provided. If channel expression is not affected in *Ctr1^-/-^* cells, why is there a two fold difference in current intensity following TTM treatment between WT and *Ctr1^-/-^* cells assuming that TTM can chelate all the Cu^++^ in the cells?

7) The structural constraints related to Cu^++^ coordinating four His358 residues should be discussed. What are the minimal distances between adjacent imidazole nitrogen atoms to insure Cu^++^ coordination? How does the proposed structural model for the channel closed state impact on the possibility of a dimer-dimer structure for the channel open state? The model presented in Figure 5 presumes that this dimer-dimer structure does not prevail in the channel open state configuration.

8) The increase in KCa3.1 channel activity in *Ctr1^-/+^* CD4 T cells was predicted by the patch clamping studies. However, the increase in His358 phosphorylation was not predicted. This finding raises questions concerning mechanism. Is His358 phosphorylation a passive response to lack of copper binding to His358? This response differs from the major conclusion of this study – that phosphorylation prevents copper interactions with KCa3.1. The observation suggests a more complicated relationship between copper binding and histidine phosphorylation than the authors' conclusions.

9) Very limited information is provided concerning statistical analysis. For example, bar graphs and error bars are presented in Figure 4, but it is unclear whether these data represent mean {plus minus} SEM or something else, and the number of biological replicates is unclear. What statistical test was performed? Are differences statistically significant? This information should be presented for all figure panels.

---

## [Author Response]

*1) Copper binding to His358 is inferred from indirect studies, but is not directly demonstrated – this should be acknowledged. The dose response of copper (Figure 1) shows that 1µM had no effect, 10µM shows an inhibitory effect, and that 100µM was saturating. How does this dose response compare with physiological intracellular copper concentrations?*

We have acknowledged in the revised Discussion that we do not have direct evidence of copper binding to His358.

We typically do our patch clamping in the presence of EGTA to buffer cytosolic and bath calcium. As pointed out by the reviewers, EGTA is also a potent copper chelator. We have now repeated the inside/out (I/O) patch clamp experiments without EGTA (Figure 2) so as to more accurately determine the lowest free copper concentration that inhibits. These experiments demonstrated that 100nm CuCl_2_ inhibits KCa3.1, which is a 100-fold lower than with EGTA present.

Most of the intracellular copper is sequestered, and free intracellular copper levels are very low. This has led to much speculation as to how copper metalloenzymes become loaded with copper. Some of the ideas include copper metallochaperones that function to load copper onto enzymes and presumably KCa3.1. Thus, it is very difficult, and likely not relevant, to extrapolate the amount of copper needed to inhibit in vitro (I/O patches) to intracellular copper concentrations.

*2) The concentration of free Cu^++^in the solutions used for the whole cell and inside-out patch clamp experiments is a concern. The expected Cu^++^concentration in the pipette solution used in whole cell experiments and in the bath solution for the inside-out experiments should be specified. EDTA has a binding affinity of 10-18M for Cu^++^and it is likely to be the same for EGTA. Under such conditions, both solutions should be Cu^++^free and yet channel activity appears to be low in the whole cell and inside-out recordings presented in Figure 1 and Figure 2 in control conditions.*

This is an excellent point raised by the reviewers. See response #1 above.

*3) Figure 2 shows that channel activity remains high after TTM removal in 300nm Ca^2+^ (the effect of TTM is not reversible) consistent with the bathing solution being Cu^++^free and TTM being capable of chelating Cu^++^from the Cu^++^-H358 complex. However, channel activity appears low in 300nm Ca^2+^ after TTM treatment in 30nm Ca^2+^. One would have expected channel activity to be high in this case if TTM had succeeded to chelate Cu^++^from the Cu^++^-H358 complex in 30nm Ca^2+^ conditions. The authors need to discuss how the action of TTM might be modulated by Ca^2+^ and may involve more than chelating Cu^++^.*

This is another excellent point raised by the reviewers, which we missed. We have now repeated the same experiment with NDPK-B and found that treatment with NDPK-B (and GTP) followed by high calcium also does not activate KCa3.1. Rather, calcium needs to be present simultaneously for both TTM and NDPK-B to activate. This makes perfect physiological sense as this requirement would prevent aberrant activation of the channel under sub-threshold stimuli. We hypothesize that a calcium/calmodulin-induced conformational change in the channel provides access to His358 for phosphorylation by NDPK-B. We have modified our model to incorporate this finding, which is presented in the new Figure 6 and in the Discussion.

*4) Segments of the inside-out recording in Figure 2 should also be presented at a higher time resolution to compare single channel activity at 300nm Ca^2+^ + low Cu^++^, 10μM CuCl_2_ and TTM conditions. The authors also need to comment on the finding that in Figure 2, channel activity appears greater in CuCl_2_ conditions than in control before TTM application in 300nm Ca^2+^ conditions.*

The data shown in Figure 2 and Figure 3 are mean patch current. The single channel activity will not be apparent with higher time resolution due to the existence of many channels in each patch. If the reviewers would like, we can also present the results as mean patch in the form of a bar graph for additional clarity.

High calcium results in activation of other potassium channels in the I/O patch that are not KCa3.1, which is evident from the failure of the KCa3.1 inhibitor to block this current. This finding provides further evidence that copper specifically inhibits KCa3.1, as copper only inhibits the TRAM-34-inhibitable current (KCa3.1) and not the non-KCa3.1 current induced by calcium. This is not evident in the other tracings, as baseline current shown is in 300nm calcium, not 30nm, as in the old Figure 2 and the new Figure 3.

*5) The proposed mechanism implies that Cu^++^coordination by four H358 is state dependent, with Cu^++^binding taking place in the channel closed configuration. Accordingly, the potency of CuCl_2_ to decrease channel activity following TTM treatment should be Ca^2+^ dependent with CuCl_2_ being less potent in saturating Ca^2+^ conditions (10μM Ca^2+^). The Ca^2+^ dependency of CuCl_2_ mediated inhibition after TTM treatment should be presented in inside-out experiments to confirm state dependency.*

We have now performed this experiment and indeed found that copper inhibition of KCa3.1 is dependent on calcium levels, with a 10-fold higher concentration of copper (1μM) required to inhibit the channel under saturating calcium concentrations (new Figure 2). These results have been incorporated into our model (see Discussion and the new Figure 6).

*6) The significance of the results illustrated in Figure 3 would be clearer if information on the mechanisms underlying Cu^++^cellular homeostasis, and on the expected basal Cu^++^intracellular level is provided. If channel expression is not affected in Ctr1^-/-^ cells, why is there a two fold difference in current intensity following TTM treatment between WT and Ctr1^-/-^ cells assuming that TTM can chelate all the Cu^++^in the cells?*

See response #1 above with respect to free copper levels in cells. We think that the difference in activity is due to the failure of TTM (applied exogenously) to chelate all of the copper bound to KCa3.1 in WT MEFs. In fact, this is consistent with our findings that calcium binding to calmodulin is required for TTM to chelate copper away from KCa3.1 (new Figure 3). Since intracellular calcium concentrations are unlikely to be saturating, TTM should not be able to chelate all of the channel-bound copper.

*7) The structural constraints related to Cu^++^coordinating four His358 residues should be discussed. What are the minimal distances between adjacent imidazole nitrogen atoms to insure Cu^++^coordination? How does the proposed structural model for the channel closed state impact on the possibility of a dimer-dimer structure for the channel open state? The model presented in Figure 5 presumes that this dimer-dimer structure does not prevail in the channel open state configuration.*

As mentioned in the previous version of the manuscript, we have unpublished crystallographic data showing that the C-terminal region of KCa3.1 forms a four-helix bundle. Suggestively, in this structure, His389 (31 residues C-terminal to the phosphorylation site, His358) occupies an inward-facing ‘a’ position in the heptad repeat, and the four copies of His389 (via N3 in the imidazole ring), along with an axial water molecule, bind a Cu(II) ion on the four-fold axis. (CuCl_2_ was added prior to crystallization trials.) The N3 (imidazole ring)-Cu(II) distance is 2.0Å (C4 symmetry–all four the same), and the O (H_2_O)-Cu(II) distance is 2.4Å. This structure demonstrates that four histidine residues in the ‘a’ position of a four-helix bundle (in a homotetramer) are capable of binding a Cu(II) ion on the four-fold axis. Based on coiled-coil prediction software (see Minor Point #5 below), our model is that His358 is also positioned in an ‘a’ position in a four-helix bundle, coordinating a Cu(II) ion on the four-fold axis similar to His389.

Because our crystal structure of the C-terminal four-helix bundle is unpublished, and because of the confusion that could arise by showing copper coordinated by His389 instead of His358 (the subject of the present study), we prefer to simply state in the Discussion that His358 is predicted to occupy an inward-facing ‘a’ position, and that the four copies would be positioned to coordinate (via N3) a Cu(II) ion on the four-fold axis.

For several reasons, we believe that the 2:2 SK2 (KCa2.2):calmodulin (CaM) structure reported by Schumacher et al. (Nature 410, 1120 (2001)) is artifactual and that their dimer-dimer model for the open-channel state is not correct. The fundamental reason for believing their model is incorrect is that the dimer-dimer model is incompatible with the C4 symmetry of the homotetrameric SK/KCa channels. We believe that the 2:2 complex they observed in solution and in the crystal structure (with calcium present) is an artifact of the C-terminal 6xHis-tag included, which contained a heterologous leucine residue (LEHHHHHH) that interacts with the second molecule of CaM in the dimer (i.e., this might help to stabilize the 2:2 complex). Our solution data (unpublished), as well as data published by Halling et al. (J. Gen. Physiol. 143, 231 (2014)), show unequivocally a 1:1 SK2 (or SK4):CaM complex, with or without calcium present. Four 1:1 complexes can be assembled into a structure that is consistent with the C4 symmetry of the channel. Such a model was proposed by Sachyani et al. (Structure 22, 1582 (2014)) for Kv7.1, which is also activated by calcium-CaM and shows sequence similarity with the KCa channels in the CaM binding domain. We favor this model for KCa3.1-CaM in the open state, which is why this paper and not the Schumacher et al. (2001) paper is referenced in the new Figure 6 legend.

*8) The increase in KCa3.1 channel activity in Ctr1^-/+^ CD4 T cells was predicted by the patch clamping studies. However, the increase in His358 phosphorylation was not predicted. This finding raises questions concerning mechanism. Is His358 phosphorylation a passive response to lack of copper binding to His358? This response differs from the major conclusion of this study – that phosphorylation prevents copper interactions with KCa3.1. The observation suggests a more complicated relationship between copper binding and histidine phosphorylation than the authors' conclusions.*

We have now addressed these points in the model presented in the Discussion and in the new Figure 6.

*9) Very limited information is provided concerning statistical analysis. For example, bar graphs and error bars are presented in Figure 4, but it is unclear whether these data represent mean {plus minus} SEM or something else, and the number of biological replicates is unclear. What statistical test was performed? Are differences statistically significant? This information should be presented for all figure panels.*

Detailed statistical methods along with the significance levels have been added to figure legends where indicated.